# Trends and socioeconomic inequalities of recommended antenatal care services utilization in Ethiopia: A decomposition analysis using Ethiopian nationwide Demographic Health Surveys 2011–2019

**Yawkal Tsega**[ID][1]*, **Abel Endawkie**[ID][2], **Gebeyehu Tsega**[ID][3], **Asnakew Molla Mekonen**[1], **Yeshimebet Ali Dawed**[4], **Chad Stecher**[ID][5]

1 Department of Health System and Management, School of Public Health, College of Medicine and Health Sciences, Wollo University, Dessie, Ethiopia, 2 Department of Epidemiology and Biostatistics, School of Public Health, College of Medicine and Health Sciences, Wollo University, Dessie, Ethiopia, 3 Department of Health Systems Management and Health Economics, School of Public Health, College of Medicine and Health Sciences, Bahir Dar University, Bahir Dar, Ethiopia, 4 Department of Public Health Nutrition and Dietetics, School of Public Health, College of Medicine and Health Sciences, Wollo University, Dessie, Ethiopia, 5 College of Health Solutions, Arizona State University, Phoenix, Arizona, United States of America

* yawkaltsega@gmail.com

## Abstract

### Background

Antenatal care (ANC) services are essential to reduce maternal and newborn morbidity and mortality rates. However, the trends and socioeconomic inequality of utilizing recommended ANC services has not been well studied in Ethiopia. Therefore, this study aims to investigate the trends and socioeconomic disparities in receiving recommended ANC services among Ethiopian women.

### Methods

This study used recent Ethiopian Demographic Health Surveys (EDHS) conducted in 2011, 2016, and 2019. Binary logistic regression model was employed to assess the association between receiving the recommended ANC services and explanatory variables and socioeconomic disparities were estimated through concentration index (CIX) analysis. Moreover, Wagstaff approach was used to decompose the relative CIX to the contribution of explanatory variables for the observed disparities.

### Results

This study found that 37.37% (95%CI: 36.46–38.28%) of mothers utilized the recommended ANC services in Ethiopia. The trend in the coverage of recommended ANC services increased from ~30% in 2011 to 44.70% in 2019. Mother's age and education, household wealth status, distance of the nearest health facility, and experiencing

**Data availability statement:** All relevant data are within the manuscript and its Supporting information files.

**Funding:** The author(s) received no specific funding for this work.

**Competing interests:** The authors have declared that no competing interests exist.

**Abbreviations:** ANC: Antenatal Care, CC: Concentration Curve, CIX: Concentration Index, LMICs: Low and Middle Income Countries, EDHS: Ethiopian Demographic Health Survey, UHC: Universal Health Coverage, WHO: World Health Organization

domestic abuse (i.e., wife beating) were significantly associated with utilization of recommended ANC services. The relative estimated CIX for wealth index, mothers education, Ethiopian administrative regions, and residence were 0.15 ($P < 0.001$), 0.14 ($P < 0.001$), 0.07($P < 0.001$), and –0.11($P < 0.001$), respectively. Wealth status of the households contributed for almost two-thirds (66.58%) of the observed disparity in recommended ANC service utilization across wealth categories.

## Conclusion

The study revealed that Ethiopian women's utilization of recommended ANC services was unequal by their socioeconomic classes, with better off women more likely to utilize the recommended ANC services than worse off women. Hence, the responsible body should improve the access and quality of antenatal care services for underprivileged women in Ethiopia.

## Introduction

Antenatal care (ANC) is a cornerstone of maternal and newborn health, providing a critical opportunity to deliver essential health services during pregnancy. These services include health education, nutritional support, screening for potential complications, and preventive measures such as various immunizations [1–3]. Effective ANC can significantly reduce maternal and neonatal morbidity and mortality by identifying and managing pregnancy related health complications early [4]. Globally, the World Health Organization (WHO) recommends a minimum of eight ANC visits to ensure comprehensive care [5].

Additionally, ensuring equitable access to high-quality maternal health services, including ANC service, is a pillar of achieving universal health coverage (UHC) by 2030 [6–9]. Recommended ANC services can prevent stillbirths and newborn deaths by ensuring optimal fetal growth and development [10,11]. However, improving maternal and reproductive health remains a challenge in LMICs, where the most vulnerable women have limited or no access to ANC, which results in the worst pregnancy and health outcomes [2].

Ethiopia has the highest burden of maternal and neonatal health problems and high socioeconomic inequality in recommended ANC services utilization across various socioeconomic groups. The latest Ethiopian Mini Demographic Health Survey (EMDHS 2019) stated that the full ANC coverage was only 43% with a maternal mortality rate of 401 deaths per 100,000 live births, far more higher than the WHO target of 70 maternal deaths per 100,000 live births [12]. Moreover, existing literature on ANC utilization in Ethiopia revealed significant improvements in coverage over the past two decades, yet substantial socioeconomic disparities persist [13,14]. Women from wealthier households, urban areas, and those with higher educational attainment are more likely to receive the recommended number of ANC visits, while poorer, less educated, and rural women are disadvantaged [3,13,14]. These inequalities contribute to higher maternal and newborn morbidity and mortality among the less privileged groups. The key drivers of these disparities include household economic status, maternal education, and place of residence [3,14].

Furthermore, studies on the utilization of recommended antenatal care (ANC) services in Ethiopia revealed significant gaps and challenges. For instance, a study in rural south-central Ethiopia found that only 76.1% of women attended at least one ANC visit, with an average of 2.5 visits, and only 2.6% attended in their first trimester [15]. Another study stated that only 40% of women had four or more ANC visits, with overall effective coverage at 12% when

adjusted for quality [16]. The timing of the first ANC booking was also problematic, with many women not starting within the first 12 weeks of gestation [17].

Despite several studies conducted on ANC services, the trends and socioeconomic inequalities in recommended ANC utilization have not been well studied in Ethiopia. Previous studies tried to assess these disparities but lacked a comprehensive analysis of the inequalities over time and determining the absolute and percentage contribution of explanatory variables [3,13,14,18]. Therefore, this study aimed to assess the trends and socioeconomic inequalities in recommended ANC service utilization among Ethiopian pregnant women. We used data from the three rounds of EDHS (2011, 2016, and 2019) to analyze trends and socioeconomic inequalities in recommended ANC services utilization. The study employed binary logistic regression model to assess associations between recommended ANC services utilization and explanatory variables, relative concentration index to estimate disparities across socioeconomic groups, and applies the Wagstaff concentration index decomposition approach [19] to identify key contributors to these observed inequalities.

## Materials and methods

### Study setting, data source, and population

Ethiopia is found in Horn Africa and its elevation ranges from 125 meters below sea level in the Afar Depression to 4550 meters above sea level in the Ras Dejen Mountains in the Amhara Region. The study used data from the three most recent rounds of the Ethiopian Demographic and Health Surveys (EDHS) conducted in 2011, 2016 and 2019. These surveys provide nationally representative data on demographic and health indicators, including birth patterns, mortality rates, maternal and child health, family planning practices, and childhood nutritional status [20,21].

The EDHS employs a two-stage cluster design, stratified in nature, where enumeration areas (EAs) selected in the first stage, and 28–30 households were chosen from each EAs in the second stage. The datasets have been made freely available on the internet for academia and researchers to use.

All pregnant women five years preceding each respective survey year were the source populations, and all pregnant women five years preceding each survey year in the selected EAs were the study populations. We excluded pregnant women who had missing information related to the outcome variable and/or any of the explanatory variables of interest. The final weighted sample size was 10,967 pregnant mothers, with 3,421, 4,638, and 2,908 mothers from the 2011, 2016, and 2019 EDHS, respectively.

### Outcome variable

The outcome variable for this study was recommended ANC (Yes/No) services utilization. It was classified as "Yes" if the mother attended her first ANC visit within first trimester of her pregnancy and if she completed at least four ANC visits, attending a sufficient number of ANC visits is crucial for the monitoring of the mother's health and the development of the fetus. Otherwise, it was classified as "No". The classification was made based on the previous literatures and Ethiopian ANC guideline [12,22].

### Socioeconomic variables and measurements

In this study we used four **individual** (maternal education level) and **community/household** level (wealth index, place of residence, and Ethiopian administrative regions) variables as a measurement of socioeconomic status of pregnant women. Wealth index was built by

considering the durable assets owned by households, household sanitation, source of drinking water, and housing conditions. Each asset was assigned a weight or factor score based on its perceived indication of wealth, and standardized to a normal distribution. A standardized score was then assigned to each household based on their possession of these assets, and the scores were summed to determine the total household score. The households were ranked based on these scores and divided into five groups, each representing 20% of the population, known as quintiles. The lowest quintile represented the poorest households, while the highest quintile represented the richest households. The use of wealth quintiles is preferred over income or consumption in assessing long-term economic status. Moreover, maternal education level was categorized into four groups: no education, primary education, secondary education, and higher education. The 11 Ethiopian administrative regions, comprising 9 regions and 2 city administrations, were ranked based on their human development index (HDI), with the region with the lowest HDI (Afar) ranked first and the region with the highest HDI (Addis Ababa) ranked last. Lastly, the type of maternal residence was classified as either urban or rural [23–25].

## Other explanatory variables

Further variables were extracted from EDHS datasets (2011–2019) through reviewing previous literatures. These variables, known for their association with recommended ANC service utilization, include factors like distance of health facilities from maternal place (categorized into two, if the distance of the nearest health facility is perceived to be the problem to access maternal and other healthcare services considered as a barrier, classified as "big problem", if otherwise "not big problem"), husband/partner education, and gender of the household head.

The classification of all extracted explanatory variables was established in alignment with insights from prior literatures [26,27]. Wife-beating was measured by using the variables burning food, arguing with husband, going out without telling husband, neglecting the children, and refusing to have sexual intercourse with her husband. If a women say "yes" at least one from the above five variables, she was considered as accepting wife-beating [28]. Moreover, media exposure was measured using three main variables: frequency of listening to the radio, watching television, and reading newspapers or magazines. These variables shared the same response options, which included "not at all", "less than once a week", and "at least once a week". Consequently, individuals who responded "not at all", "once a week", and "at least once a week" for all variables were categorized as having no exposure, limited exposure, and having exposure, respectively [29] (Table 1).

## Statistical analysis

All the statistical analyses were weighted using the survey weighting variable provided by the EDHS datasets. Descriptive statistics was made to summarize the socioeconomic and demographic characteristics of the study population, and weighted prevalence along with a 95% confidence interval was presented. The association between explanatory variables and recommended ANC services utilization was assessed using bivariable logistic regression analyses. To determine the independent association of these explanatory variables on the outcome variable (i.e., recommended ANC services utilization), confounding effect was controlled through multivariable logistic regression analysis. Inclusion in the multivariable model involved variables with a significance level of $P < 0.2$ in the bivariable logistic regression. The overall fitness of the binary logistic regression model was assessed using the Hosmer and Lemeshow test ($P = 0.100$). Adjusted Odds Ratios (AOR) with their respective 95% confidence intervals were used to measure the strength of the associations between recommended ANC services

**Table 1. List of extracted explanatory variables for recommended ANC services utilization with their respective categories from 2011–2019.**

| Variables | Category |
|---|---|
| Type of place of residence | Rural, Urban |
| Family size | 1–4, 5–6, ≥ 7 |
| Mother's age | 15–24, 25–34, 35–49 |
| Mother's education | No education, Primary, Secondary, Higher |
| Mother's current marital status | Currently unmarried*, currently married |
| Sex of household head | Male, Female |
| Age of household head | ≤ 24, 25–64, ≥ 65 |
| Wealth status of the household | Poorest, Poorer, Middle, Richer, Richest |
| Age of mother at first birth | <18, 18–24, ≥ 25 |
| Distance of health facility | Big problem, Not big problem |
| Husband/partner's occupational status | Not working, Working |
| Husband/partners education | No education, Primary, Secondary, Higher |
| Mother's occupational status | Not working, Working |
| Husband/partner's age | ≤24, 25–64, ≥ 65 |
| Birth order | ≤3, 4–6, ≥ 7 |
| Media Exposure | No, limited exposure, have exposure |
| Wife beating | Yes, No |

*single, widowed, and divorced.

utilization and explanatory variables. Statistical significance for identifying significant factors associated with recommended ANC service utilization was determined at a $P < 0.05$.

## Inequality measurement

The inequalities in recommended ANC services utilization across socioeconomic characteristics were estimated using the concentration curve (CC) and concentration index (CIX) in their relative form (without adjustment) [19,30]. The CIX used in this study indicates horizontal inequity, assuming that every mother in the study had an equal need for recommended ANC services utilization.

The construction of CC involved plotting the cumulative proportion of pregnant mothers ranked by their socioeconomic characteristics (starting from the less privileged category, except residence (urban first and rural second) on the x-axis, and the cumulative magnitude of recommended ANC services utilization on the y-axis. A perfect equality is represented by a 45-degree slope depicted from northwest to northeast.

If the CC overlaps with line of equality, it indicates that recommended ANC services utilization is equal among mothers across their socioeconomic characteristics. However, if the CC move away from the line of equality below or above, it suggests the presence of inequalities in recommended ANC services utilization, favoring mothers with high or low socioeconomic classes, respectively. The larger the distance between the CC and the line of equality, the greater the degree of inequality. The CIX is equal to twice of the area between the line of equality and CC. It takes a value between −1 and +1, where a value of 0 indicates equal distribution of recommended ANC services utilization across socioeconomic groups.

Moreover, a positive value of CIX implies that ANC services utilization is concentrated among the higher socio-economic groups (pro-privileged). On the other hand, a negative value of CIX indicates that recommended ANC service utilization is primarily concentrated at

lower socioeconomic groups (pro-under privileged). The estimation of CIX was done by using convenient covariance formula described by O'Donnell et al [19] as shown below,

$$CIX = \frac{2}{\mu}cov(h, r) \tag{1}$$

Where h is the health variable, recommended ANC services utilization in this study, μ is its mean, and r = i/N is the fractional rank of individual i in the living standards (socioeconomic variables distribution, with i = 1 for the low socioeconomic group and i = N for the highest socioeconomic group. The user-written STATA commands lorenz estimate [31] and conindex [31,32] were used to sketch CC and measure CIX, respectively.

## Decomposition of relative concentration index

The decomposition of the relative CIX was carried out to estimate important explanatory factors contribution to the total estimated socioeconomic inequalities in recommended ANC services utilization across various socioeconomic characteristics (wealth index, region, residence and mothers' education level). Wagstaff and O'Donnell [19,32,33] approach was used to decompose relative CIX. Wagstaff demonstrated that the CIX can be decomposed into the contributions of individual factors to total estimated socioeconomic related inequality [33]. The contribution of each determinant of recommended ANC service utilization to the overall socioeconomic inequality was determined by multiplying the determinant sensitivity to recommended ANC services utilization (elasticity) with the level of socioeconomic inequality associated with that determinant (CIX of the determinant). The residual component represents the portion of the CIX that cannot be explained by the determinants considered in the analysis.

## Ethical consideration

Ethical approval was not required for this study as we used the demographic and health survey, which identifies all data before making it public. We obtained an authorization letter from CSA to download the DHS dataset at https://dhsprogram.com/. All methods of the DHS data collection were performed in accordance with the relevant guidelines and regulations of Ethiopia.

## Results

### Descriptive analysis

The weighted rate of recommended ANC services utilization in Ethiopia was 37.37% (95 CI: 36.46–38.28%). The descriptive analysis shows that the majority of the population resides in rural areas (75.80%) and falls within the age range of 25–34 (51.46%). Moreover, more than half of mothers have no education (51.05%), followed by primary education (35.91%). About 84.65% of the households were headed by male, 14.66% and 26.21% of the households were classified as poorest and richest, respectively with regard to their wealth status. Likewise, 55.85% of the mothers reported that the distance of the health facility from their place was a big problem to access maternal and/or other medical healthcare services (Table 2).

### Predictors of recommended ANC services utilization

From weighted bi-variable logistic regression, about 15 variables were eligible (p < 0.2) for multivariable logistic regression analysis. In the multivariable logistic regression, mothers age and education, wealth status, distance of health facility, husband/partner education, birth

**Table 2. Weighted descriptive analysis of explanatory variables for recommended ANC service utilization in Ethiopia from 2011–2019).**

| Variables | Category | Weighted Frequency | Weighted Percent (%) |
|---|---|---|---|
| Recommended ANC visit | Yes | 4,078 | 37.37 |
| | No | 6,835 | 62.63 |
| Type of place of residence | Urban | 2,642 | 24.20 |
| | Rural | 8,271 | 75.80 |
| Family size | 1–4 | 3,785 | 34.69 |
| | 5–6 | 3,769 | 34.54 |
| | ≥ 7 | 3,359 | 30.78 |
| Mother's age | 15–24 | 2,893 | 26.51 |
| | 25–34 | 5,616 | 51.46 |
| | 35–49 | 2,404 | 22.03 |
| Mother's education | No education | 5,571 | 51.05 |
| | Primary | 3,919 | 35.91 |
| | Secondary | 917 | 8.40 |
| | Higher | 506 | 4.63 |
| Current marital status | Currently unmarried* | 622 | 5.70 |
| | Currently married | 10,291 | 94.30 |
| Mother's occupational status | Not working | 3,746 | 47.40 |
| | Working | 4,158 | 52.60 |
| Sex of household head | Male | 9,237 | 84.65 |
| | Female | 1,676 | 15.35 |
| Age of household head | ≤ 24 | 617 | 5.67 |
| | 25–64 | 9,801 | 90.02 |
| | ≥ 65 | 469 | 4.31 |
| Wealth status | Poorest | 1,599 | 14.66 |
| | Poorer | 2,089 | 19.14 |
| | Middle | 2,164 | 19.83 |
| | Richer | 2,201 | 20.17 |
| | Richest | 2,860 | 26.21 |
| Age of mother at first birth | <18 | 4,117 | 37.72 |
| | 18–24 | 5,955 | 54.57 |
| | ≥ 25 | 841 | 7.71 |
| Distance of health facility | Big problem | 4,477 | 55.85 |
| | Not big problem | 3,539 | 44.15 |
| Husband/partner's occupational status | Not working | 285 | 3.80 |
| | Working | 7,219 | 96.20 |
| Husband/partners education | No education | 1,240 | 37.79 |
| | Primary | 1,498 | 45.67 |
| | Secondary | 288 | 8.78 |
| | Higher | 255 | 7.76 |
| Husband/partner's age | ≤ 24 | 302 | 4.07 |
| | 25–64 | 6,978 | 93.86 |
| | ≥ 65 | 154 | 2.07 |
| Birth order | ≤ 3 | 6,141 | 56.27 |
| | 4–6 | 3,184 | 29.18 |
| | ≥ 7 | 1,587 | 14.55 |
| Media Exposure | Yes | 4,437 | 55.34 |
| | No | 3,581 | 44.66 |

*(Continued)*

**Table 2.** (Continued)

| Variables | Category | Weighted Frequency | Weighted Percent (%) |
|---|---|---|---|
| Wife beating | Yes | 5,145 | 65.58 |
| | No | 2,700 | 34.42 |

*single, widowed, and divorced

order and wife beating were found to be significant predictors of recommended ANC services utilization at $P < 0.05$ significance level.

The odds of recommended ANC services utilization among mothers whose age category fall between 25–34 and 35–39 were 1.69 (AOR: 1.69, CI: 1.32–2.16) and 2.77 (AOR: 2.77, CI: 1.91–4.00) times higher, respectively, compared to mothers with age range of 15–24. The likelihood of recommended ANC services utilization was 2.31 (AOR: 2.31, CI: 1.54–3.46) and 2.31(AOR: 2.31, CI: 1.37–3.91) times higher among mothers with secondary and higher educational level, respectively, compared with mothers with no education.

Moreover, mothers with their husband education level secondary and primary demonstrated a 1.38 (AOR: 1.38, CI: 1.14–1.70) and 1.63(AOR: 1.68, CI: 1.16–2.31) fold increase, respectively, in recommended ANC services utilization compared to those mothers with their husband has no education. Furthermore, richest mothers were 1.73 (AOR: 1.73, CI: 1.17–2.57) times more likely to utilize recommended ANC services than the poorest mothers. Mothers with no wife beating were 1.27 (AOR: 1.27, CI: 1.05–1.53) times more likely to utilize recommended ANC services compared with mothers having wife beating (Table 3).

## Trends of recommended ANC service utilization in Ethiopia

The proportion of women who received recommended ANC services increased from ~30% in 2011 to 38.12% in 2016 and to 44.70% in 2019 as shown in the figure below (Fig 1).

## Socioeconomic inequality in recommended ANC services utilization

More than one-third (38.44%) of mothers in the highest-wealth category had recommended ANC service utilization which is higher compared to nearest to one-tenth (10.46%) of mothers in the poorest category. The graph also shows that recommended ANC service utilization increases as wealth category move from the poorest to the richest quintile (Fig 2).

The estimated CIX indicated there is a significant disparities in recommended ANC services utilization across mothers' socioeconomic categories. The estimated CIX value of recommended ANC services utilization as ranked by their wealth status was 0.15 (95%CI: 0.13–0.17) with a significant $P < 0.001$. A positive CIX value suggests a more noticeable concentration of recommended ANC services utilization among richest mothers compared to poorest mothers (Fig 3). Similarly, the study showed that there is a significant inequality of recommended ANC services utilization across mothers' educational level category. The estimated CIX value of recommended ANC services utilization as ranked by their educational level was 0.14 (95%CI: 0.11–0.16), the positive CIX revealed that educated mothers utilized recommended ANC services than the non-educated mothers (pro-educated), with a significant $P < 0.001$ (Fig 4) (Table 4).

Additionally, this study uncovered there is a significant inequality in recommended ANC service utilization across Ethiopian administrative regions (CIX: 0.07, 95%CI: 0.11, 0.16), as they are ranked based on their level of development, least developed first (Afar) and highly

**Table 3.  Bivariable and multivariable logistic regression for recommended ANC services utilization and its determinants among mothers in Ethiopia from 2011–2019.**

| Variables | Category | Recommended ANC visit | | COR (95%CI) | AOR (95%CI) |
|---|---|---|---|---|---|
| | | **Yes** | **No** | | |
| Type of place of residence | Urban | 1,442 | 1,200 | 2.57 (2.35–2.81) | 1.03 (0.76–1.40) |
| | Rural | 2,636 | 5,635 | 1 | 1 |
| Family size | 1–4 | 1,603 | 2,182 | 1.60 (1.45–1.76) | 1.19 (0.90–1.57) |
| | 5–6 | 1,417 | 2,352 | 1.31 (1.19–1.44) | 0.93 (0.74–1.18) |
| | ≥ 7 | 1,058 | 2,301 | 1 | 1 |
| Mother's age | 15–24 | 1,008 | 1,885 | 1 | 1 |
| | 25–34 | 2,185 | 3,431 | 1.19 (1.08–1.31) | **1.69 (1.32–2.16)*** |
| | 35–49 | 885 | 1,519 | 1.09 (0.97–1.22) | **2.77 (1.91–4.00)*** |
| Mother's education | No education | 1,644 | 3,927 | 1 | 1 |
| | Primary | 1,556 | 2,364 | 1.57 (1.44–1.71) | 1.16 (0.94–1.42) |
| | Secondary | 532 | 385 | 3.29 (2.85–3.80) | **2.31 (1.54–3.46)*** |
| | Higher | 346 | 159 | 5.18 (4.26–6.30) | **2.31 (1.37–3.91)*** |
| Mother's occupational status | Not working | 1,226 | 2,520 | 1 | 1 |
| | Working | 1,504 | 2,654 | 1.16 (1.06–1.28) | 1.09 (0.91–1.30) |
| Sex of household head | Male | 3,372 | 5,866 | 1 | 1 |
| | Female | 706 | 969 | 1.27 (1.14–1.41) | 1.15 (0.88–1.50) |
| Age of household head | ≤ 24 | 215 | 403 | 1 | 1 |
| | 25–64 | 3,665 | 6,136 | 1.12 (0.94–1.33) | 1.17 (0.73–1.85) |
| | ≥ 65 | 184 | 285 | 1.21 (0.95–1.56) | 1.86 (0.95–3.67) |
| Wealth status | Poorest | 427 | 1,173 | 1 | 1 |
| | Poorer | 617 | 1,471 | 1.15 (1.00–1.33) | 1.02 (0.73–1.43) |
| | Middle | 666 | 1,498 | 1.22 (1.06–1.41) | 1.10 (0.80–1.53) |
| | Richer | 800 | 1,401 | 1.57 (1.36–1.81) | 1.27 (0.92–1.76) |
| | Richest | 1,568 | 1,292 | 3.33 (2.92–3.81) | **1.73 (1.17–2.57)*** |
| Age of mother at first birth | < 18 | 1,391 | 2,726 | 1 | 1 |
| | 18–24 | 2,248 | 3,707 | 1.19 (1.09–1.29) | 1.00 (0.82–1.20) |
| | ≥ 25 | 439 | 402 | 2.14 (1.84–2.48) | 0.99 (0.67–1.46) |
| Distance of health facility | Big problem | 1,329 | 3,148 | 1 | 1 |
| | Not big problem | 1,454 | 2,085 | 1.65 (1.51–1.81) | **1.23 (1.02–1.50)*** |
| Husband/partners education | No education | 263 | 977 | 1 | 1 |
| | Primary | 454 | 1,044 | 1.62 (1.36–1.93) | **1.38 (1.14–1.70)*** |
| | Secondary | 132 | 156 | 3.15 (2.41–4.18) | **1.63 (1.16–2.31)*** |
| | Higher | 123 | 132 | 3.48 (2.63–4.61) | 1.20 (0.80–1.82) |
| Husband/partner's age | ≤ 24 | 88 | 214 | 1 | 1 |
| | 25–64 | 2,425 | 4,553 | 1.30 (1.01–1.67) | 0.77 (0.44–1.32) |
| | ≥ 65 | 45 | 109 | 1.02 (0.67–1.56) | 0.89 (0.40–2.02) |
| Birth order | ≤ 3 | 2,532 | 3,609 | 1.66 (1.47–1.87) | **2.36 (1.55–3.58)*** |
| | 4–6 | 1,074 | 2,110 | 1.20 (1.06–1.37) | **2.17 (1.55–3.03)*** |
| | ≥ 7 | 472 | 1,116 | 1 | 1 |
| Media Exposure | Yes | 1,761 | 2,676 | 1.64 (1.49–1.80) | 1.20 (0.96–1.50) |
| | No | 1,025 | 2,556 | 1 | 1 |
| Wife beating | Yes | 1,620 | 3,524 | 1 | 1 |
| | No | 1,121 | 1,580 | 1.54 (1.40–1.70) | **1.27 (1.05–1.53)*** |

*means significant at *P* < 0.05, ANC: Antenatal care, COR: Crude Odds Ratio (the output of bivariable logistic regression, one dependent variable with one independent variable), bold: significant category.

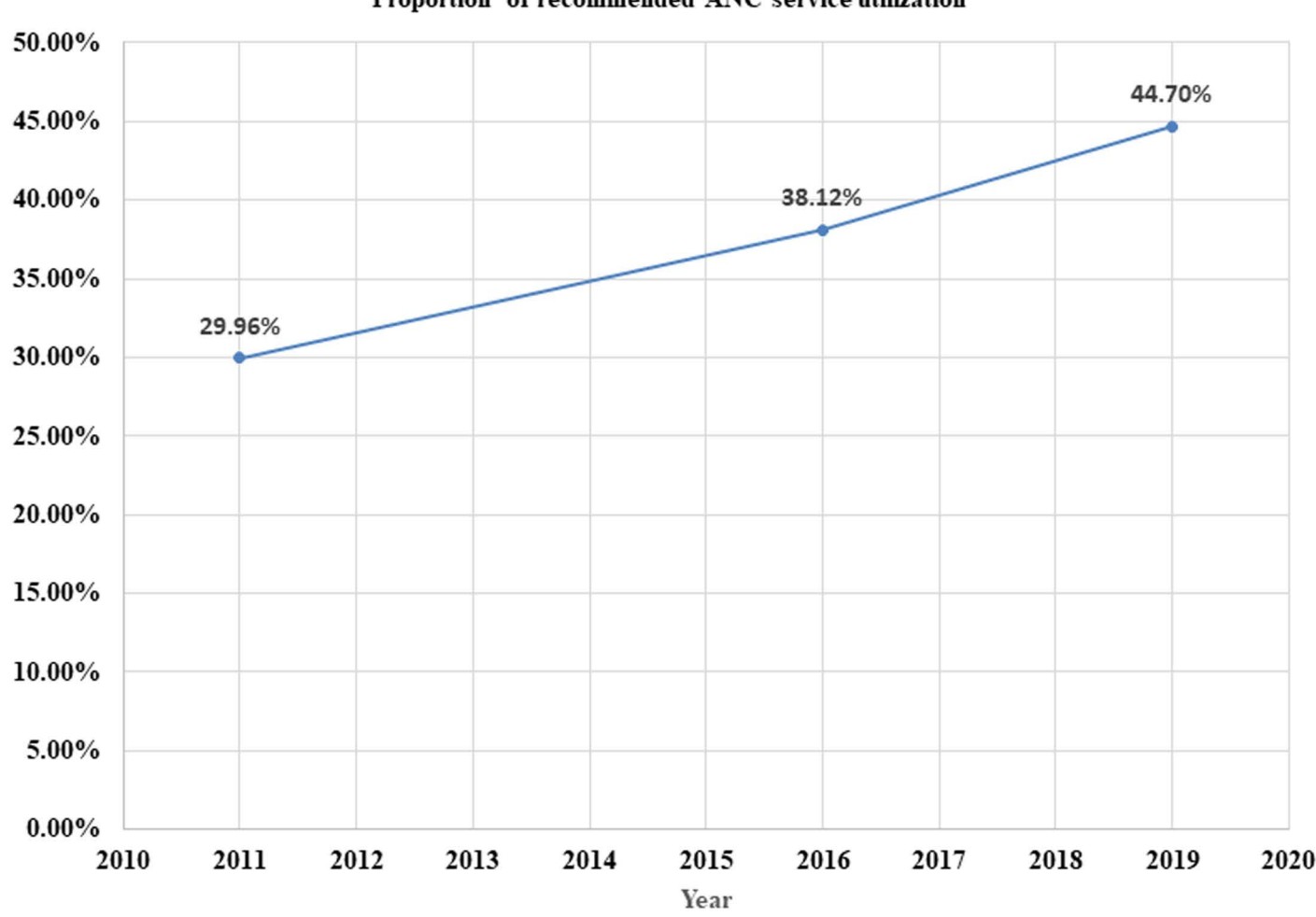

**Fig 1. Trends in the magnitude of recommended ANC service utilization in Ethiopia from 2011–2019.**

developed last (Addis Ababa), and type of place of residence (CIX: −0.11, 95%CI: −0.13, −0.09), the positive CIX revealed that mothers living in developed regions utilized recommended ANC services than the mothers living in under developed regions (pro-developed), with a significant $P < 0.001$ (Table 4).

The CC positioned below the line of equality reveals a higher concentration of recommended ANC service utilization among the richest mothers (i.e., recommended ANC services utilization is pro-rich) (**Fig 3**).

**Fig 4** depicts the concentration curve for recommended ANC services utilization as mothers ranked by their respective educational level, which reveals the CC is outlined below the line of equality, the recommended ANC services utilization is more concentrated among the educated mothers than the non-educated mothers.

**Fig 5** provides an overview of the distribution of recommended ANC service utilization across the 11 administrative regions of Ethiopia. The coverage of recommended ANC services utilization varied across these regions, ranging from 19.38% in the Somali region to 80.14% in the Addis Ababa city administration.

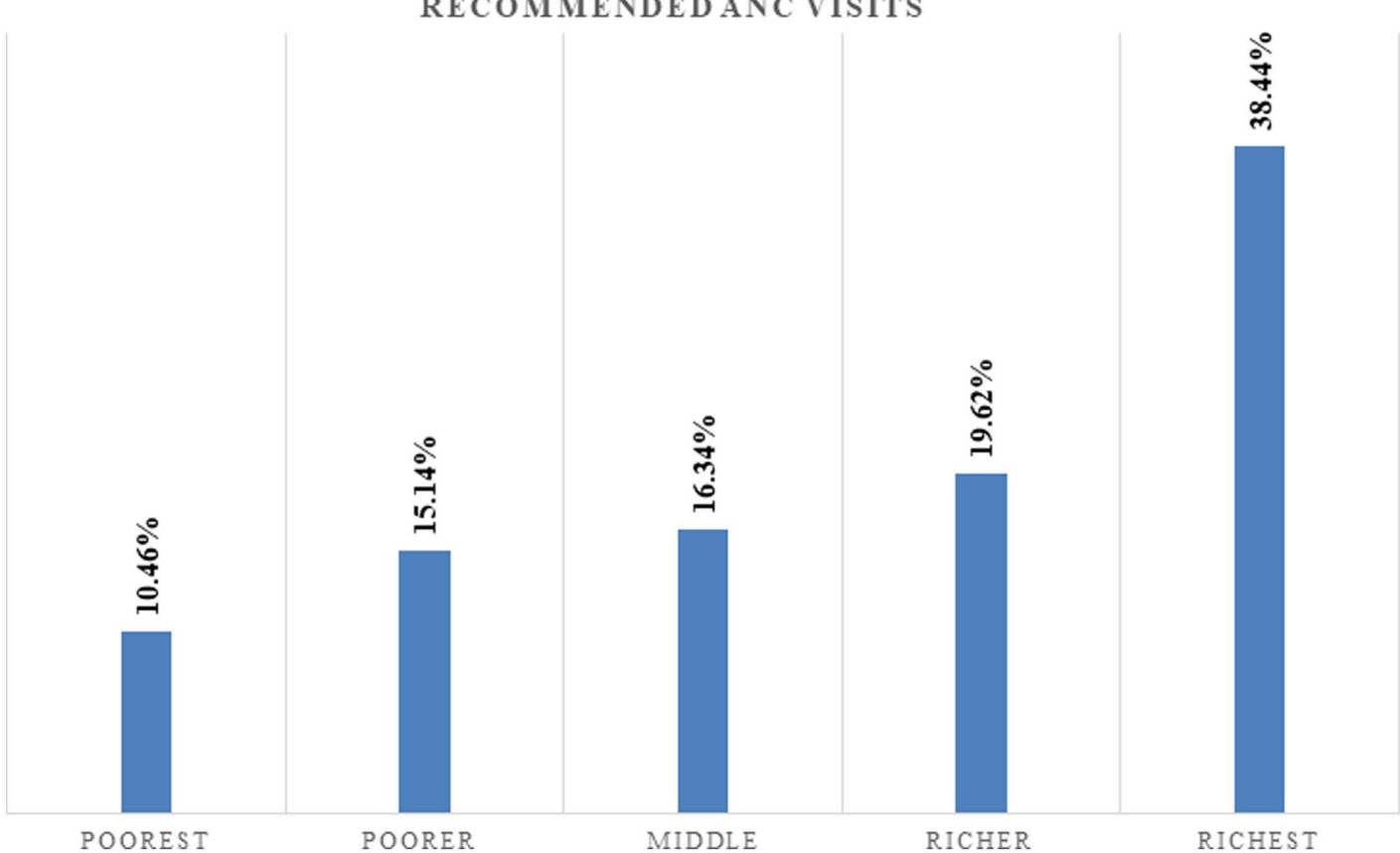

**Fig 2. Magnitude of recommended ANC service utilization across wealth categories of mothers in Ethiopia from 2011–2019.**

**Fig 6** depicts the concentration curve for recommended ANC services utilization as mothers ranked by their respective administrative region, which reveals the CC is outlined below the line of equality, the recommended ANC services utilization is more concentrated among the mothers living in developed regions than mothers living in under developed regions.

**Fig 7** depicts the concentration curve for recommended ANC services utilization as mothers ranked by their place of residence, which reveals the CC is outlined above the line of equality, the recommended ANC services utilization is more concentrated among the mothers residing in urban areas (pro-urban) than mothers residing in rural areas.

## Decomposing CIX of wealth index

The **Table 5** below states the relative contribution of each explanatory variable to the overall estimated CIX. Each variable's contribution is depicted as a percentage and absolute contributions, where a negative percentage implies a reduction in observed inequality, while a positive percentage signifies an increase in observed inequality. In the observed CIX attributed to wealth status (0.15), contributions were as follows: household wealth status accounted for 66.58% (i.e., contributed the lion share of the observed inequality), mother's education level contributed 39.41%, mothers' age contributed −2.84%, wife beating

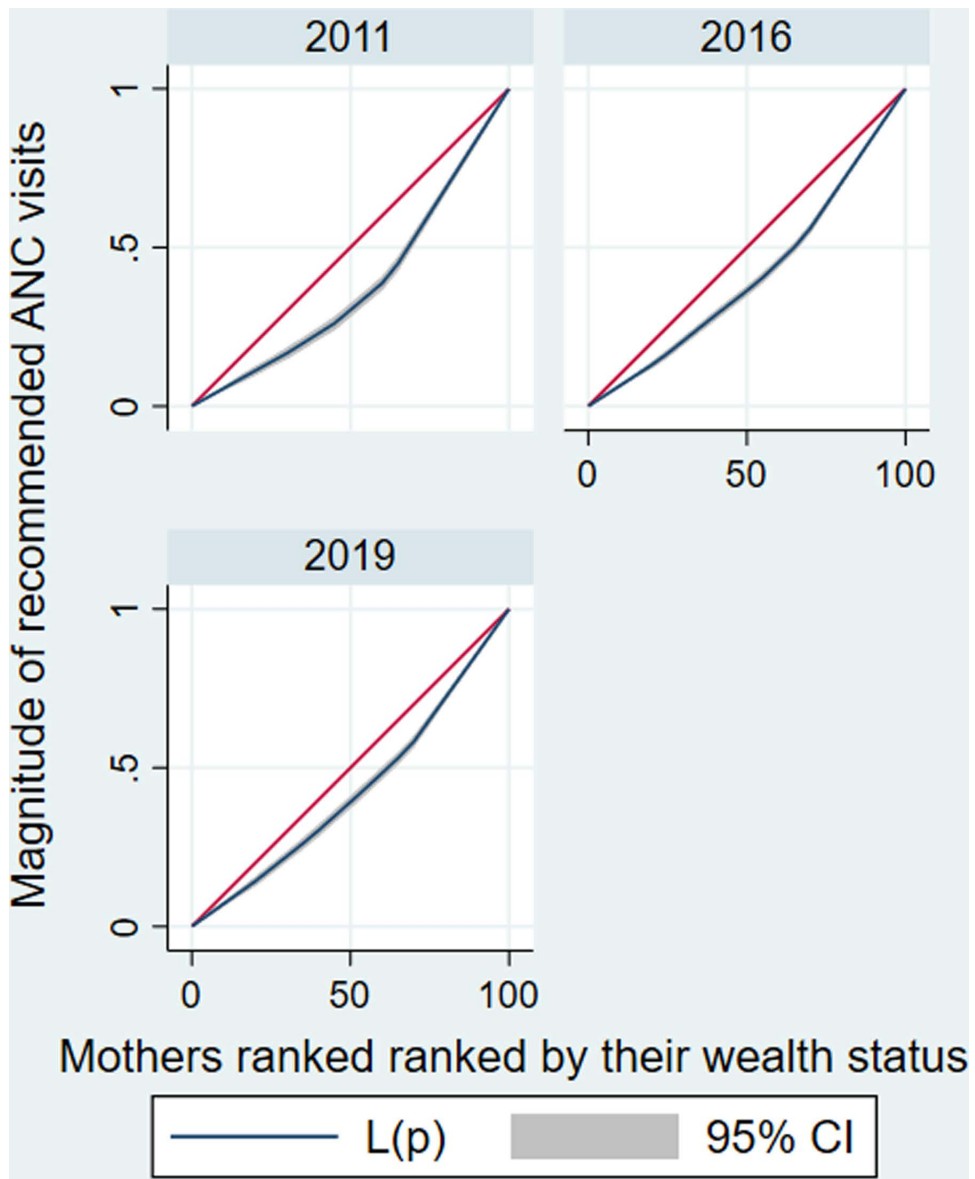

**Fig 3. Concentration curves for socioeconomic inequality (wealth status as a ranking variable) in recommended ANC service utilization in Ethiopia from 2011–2019.**

contributed 8.53%, and the distance of the health facility from the mother's place contributed 8.48% (Table 5).

## Decomposing concentration index of mothers' education

Table 6 presents explanatory variables contributing to the overall estimated CIX of maternal education (CIX: 0.14). Birth order (31.98%) and administrative region (10.84%) emerged as significant contributors to the increment of maternal education inequality in recommended ANC services utilization. However, maternal age (−25.86%) and household wealth status (−24.20%) were identified as variables contributing to the decrease in observed inequality in recommended ANC services utilization.

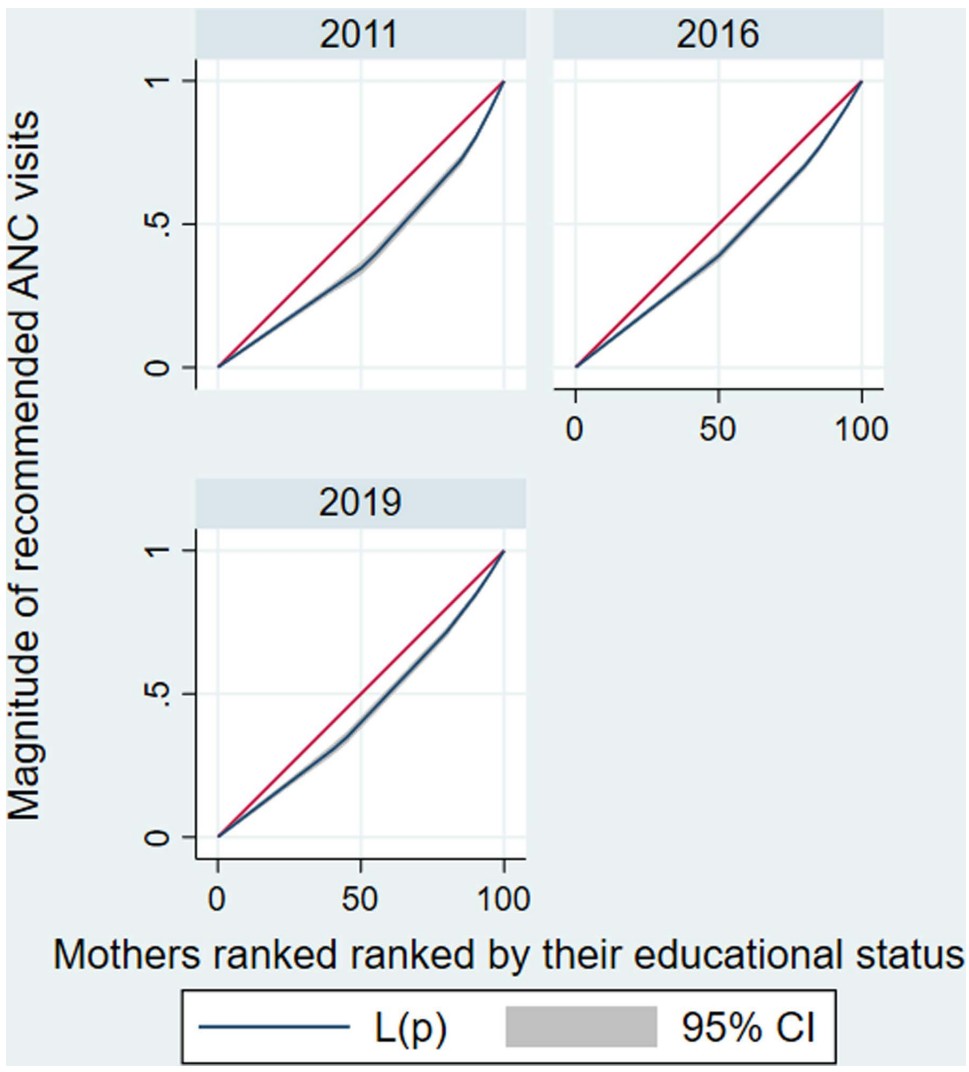

**Fig 4. Concentration curves for socioeconomic inequality (mothers' education as a ranking variable) in recommended ANC service utilization in Ethiopia from 2011–2019.**

**Table 4. Estimated concentration index for the study of trends and socioeconomic inequalities in recommended ANC service utilization in Ethiopia using EDHS 2011–2019.**

| Recommended ANC visits by | Observation | CIX | P-value | 95%CI |
|---|---|---|---|---|
| Wealth status | 10,967 | 0.15 | 0.000 | (0.13, 0.17) |
| Mother's education level | 10,967 | 0.14 | 0.000 | (0.11, 0.16) |
| Administrative region | 9882 | 0.07 | 0.000 | (0.05, 0.08) |
| Place pf residence | 10,967 | −0.11 | 0.000 | (−0.13, −0.09) |

CIX: Concentration index, ANC: Antenatal care

## Decomposing CIX of administrative region

For the CIX observed in Ethiopia's 11 administrative regions, which is 0.07, household wealth status contributed 8.86%, the region itself contributed 113.73% (thus representing the

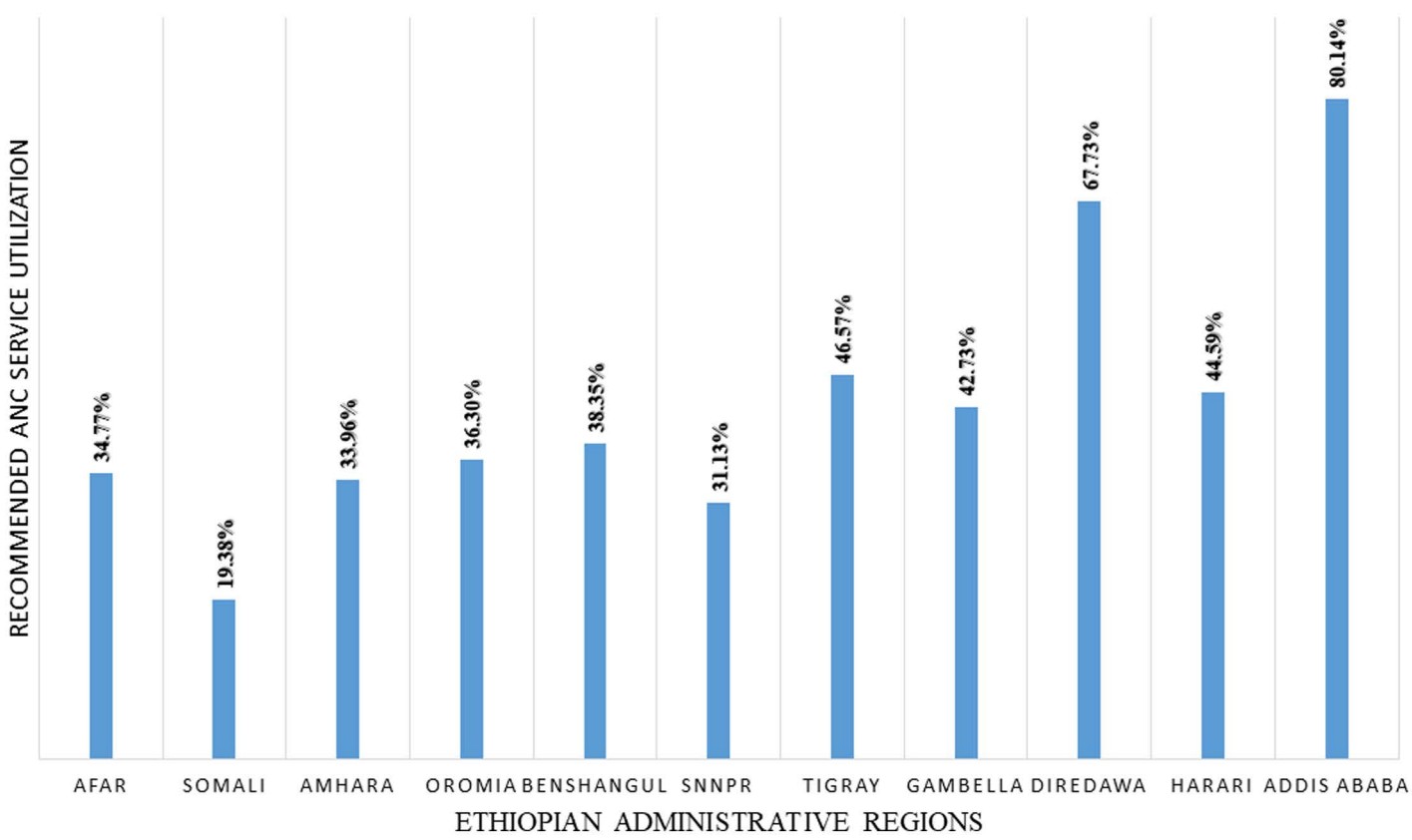

**Fig 5. Magnitude of recommended ANC service utilization across Ethiopian administrative regions from 2011–2019.**

majority of the observed disparity), the education level of the mother accounted for 25.33%, the age of the mother contributed 2.13%, wife beating accounted for 0.85%, and the distance of health facilities to the mother's place had a negative contribution of −2.13% (Table 7).

## Decomposing CIX of residence

The following table provides a visual representation of the proportional impact of different explanatory variables on the overall estimated CIX (−0.11) for the place of residence. The two variables that made the largest contributions were household wealth status, accounting for 34.40%, and mothers' education, which accounted for 31.61%. Both household wealth status and mothers' education contributed to an increase in the inequality of recommended ANC services utilization across type of place of residence of mothers (Table 8).

## Discussion

This study aimed to assess trends and socioeconomic inequalities in recommended ANC services utilization in Ethiopia using nationwide demographic health survey from 2011–2019. The study revealed that coverage of recommended ANC services utilization in Ethiopia was found to be 37.37% (95 CI: 36.48–38.28%). The result of the study suggests that ANC service utilization in Ethiopia is low and unequal across different socioeconomic groups. The findings of similar previous studies were aligned with this finding, in which ANC service in low and middle income countries are lower and unequal across several socioeconomic categories [2].

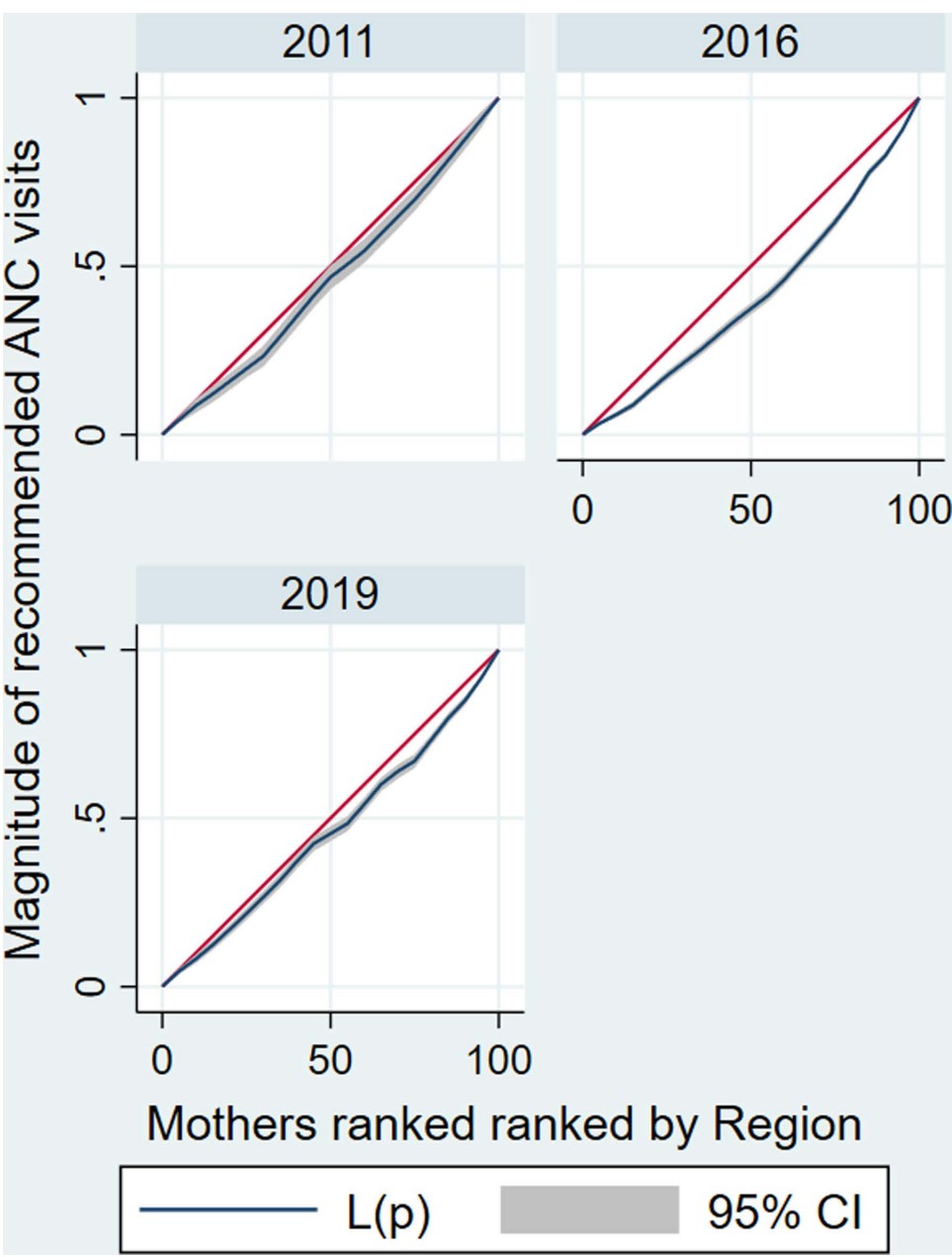

**Fig 6. Concentration curves for socioeconomic inequality (Ethiopian administrative regions as a ranking variable) in recommended ANC service utilization in Ethiopia from 2011–2019.**

The practical and policy implications of the findings are there is a need to improve the accessibility, availability, and quality of ANC services, especially in rural areas and among the poorest households, a need to address the barriers and facilitators of ANC service utilization, such as education, awareness, and cultural norms, and a need to monitor and evaluate the progress and impact of ANC service utilization on maternal and child health outcomes to ensure effective and efficient delivery of ANC services [29,34].

Furthermore, we found that women with her husband having higher education level had higher odds (AOR: 4.40) of utilizing recommended ANC services compared to those with

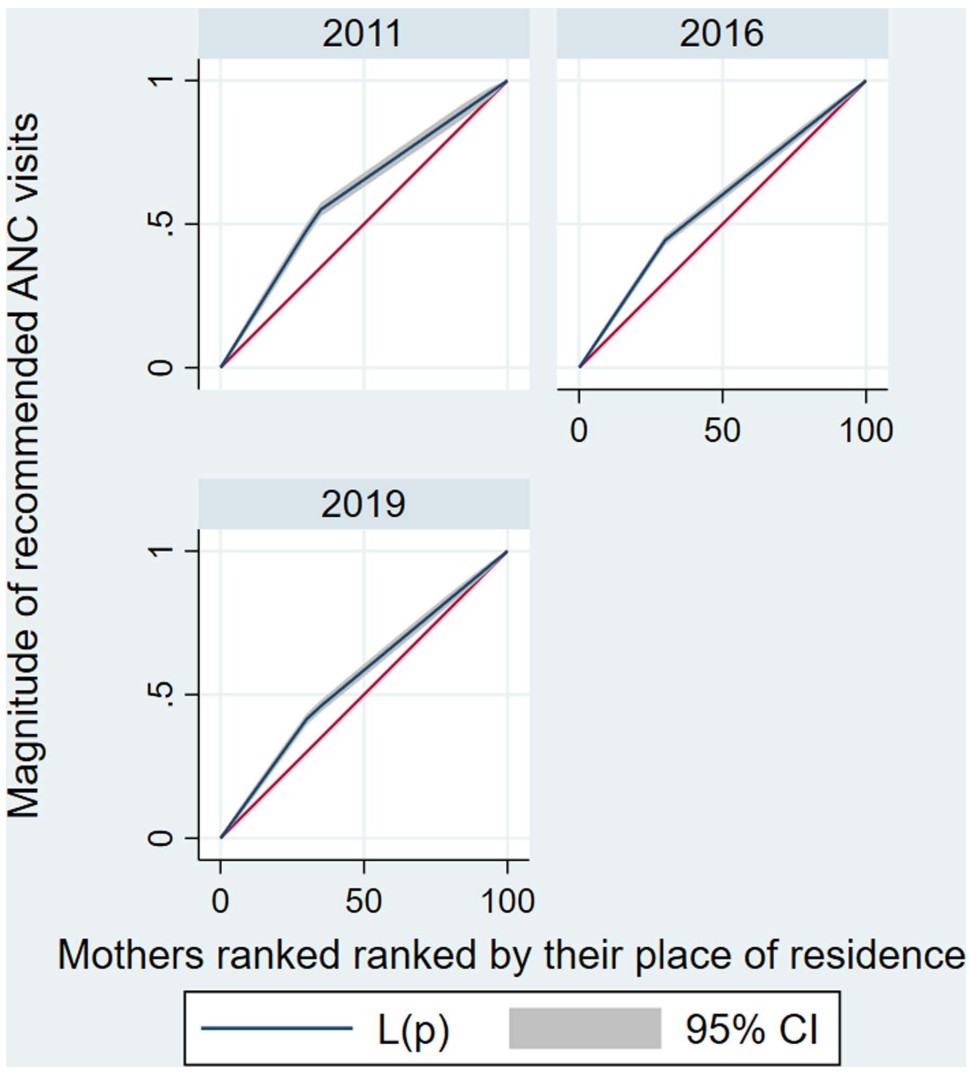

**Fig 7. Concentration curves for socioeconomic inequality (place of residence as a ranking variable) in recommended ANC service utilization in Ethiopia from 2011–2019.**

lower education levels. This link between a husband level of education and recommended ANC services utilization appears to straightforward, as individuals with higher education levels tend to possess greater awareness of personal health matters, exhibit higher self-efficacy, engage in self-care practices, and understand the significance of recommended ANC services utilization to prevent pregnancy related complications [35]. The implication is supported by the study conducted in Sub-Saharan Africa from 2008 to 2018 [36]. Additionally, educational inequality estimation between recommended ANC services utilization and mothers' level of education, supported this finding, in which recommended ANC services utilization was more concentrated among higher educated mothers.

This study also revealed there is a wealth related socioeconomic inequalities in recommended ANC services utilization in Ethiopia. The CIX for wealth status was estimated to be 0.15, indicating a significant disparity in recommended ANC services utilization across different wealth quintiles. This finding is supported by various previous studies in Ethiopia, Sub-Saharan Africa and low income countries which revealed there was wealth related

**Table 5. Decomposition of CIX for measuring contributions of explanatory variables for socioeconomic inequalities in utilizing recommended ANC services in Ethiopia from 2011–2019.**

| Variables | Category | Elasticity | CIX | Contribution to overall [CIX = 0.15, $P<0.0001$] | |
|---|---|---|---|---|---|
| | | | | **Absolute contribution** | **Percentage contribution** |
| Mother's age | 15–24(ref) | | | | |
| | 25–34 | 0.180 | 0.035 | 0.006 | 4.21 |
| | 35–49 | 0.154 | −0.069 | −0.011 | −7.10 |
| | Subtotal | | | −0.005 | −2.89 |
| Mother's education | No education(ref) | | | | |
| | Primary | 0.076 | 0.084 | 0.006 | 4.24 |
| | Secondary | 0.059 | 0.481 | 0.028 | 18.87 |
| | Higher | 0.037 | 0.661 | 0.024 | 16.30 |
| | Subtotal | | | 0.058 | 39.41 |
| Husband education | No education(ref) | | | | |
| | Primary | 0.091 | 0.017 | 0.002 | 1.07 |
| | Secondary | 0.024 | 0.459 | 0.011 | 7.39 |
| | Higher | −0.001 | 0.587 | −0.001 | −0.645 |
| | Subtotal | | | 0.012 | 7.82 |
| Wealth status | Poorest(ref) | | | | |
| | Poorer | −0.000 | −0.515 | 0.000 | 0.04 |
| | Middle | 0.014 | −0.126 | −0.002 | −1.14 |
| | Richer | 0.033 | 0.274 | 0.009 | 5.98 |
| | Richest | 0.125 | 0.738 | 0.092 | 61.70 |
| | Subtotal | | | 0.099 | 66.58 |
| Distance of health facility | Big problem(ref) | | | | |
| | Not big problem | 0.065 | 0.197 | 0.013 | 8.48 |
| | Subtotal | | | 0.013 | 8.48 |
| Birth order | ≤3 | 0.357 | 0.107 | 0.038 | 25.48 |
| | 4–6 | 0.139 | −0.126 | −0.017 | −11.67 |
| | ≥7(ref) | | | | |
| | Subtotal | | | 0.021 | 13.81 |
| Wife beating | Yes (ref) | | | | |
| | No | 0.070 | 0.182 | .013 | 8.53 |
| | Subtotal | | | 0.013 | 8.53 |
| Explained CIX | | | | 0.211 | 141.74 |
| Residual CIX | | | | −0.061 | −41.74 |

ref: reference category, CIX: Concentration index

inequality in recommended ANC services utilization [2,18,36]. The policy implications of this finding emphasize the importance of targeted interventions, financial support, awareness programs, and improvements in healthcare infrastructure to address the significant socioeconomic inequalities in recommended ANC services utilization in Ethiopia. By implementing these policies, health decision-makers can work towards achieving more equitable access to ANC services and improving maternal and neonatal health outcomes for women from lower socioeconomic classes.

Furthermore, the decomposition analysis of the CIX suggested that the highest contribution was made by wealth status (66.58%) of households for the socioeconomic inequality. It might be due to the fact that low-income households usually spend the highest share of their

**Table 6. Decomposition of CIX for measuring contributions of explanatory variables for the inequalities in recommended ANC services utilization (using mothers' educational as ranking variable) in Ethiopia from 2011–2019.**

| Variables | Elasticity | CIX | Contribution to overall [CIX = 0.14, $P < 0.0001$] | |
|---|---|---|---|---|
| | | | Absolute contribution | Percentage contribution |
| Mother's age | 0.333 | −0.105 | −0.035 | −25.86 |
| Mother education | 0.161 | 0.612 | 0.098 | 72.40 |
| Husband education | 0.037 | −0.290 | 0.011 | 7.96 |
| Wealth status | −0.353 | −0.093 | 0.033 | −24.20 |
| Distance HF | 0.242 | 0.033 | 0.008 | 5.93 |
| Birth order | −0.166 | −0.262 | 0.043 | 31.98 |
| Wife beating | 0.050 | 0.166 | 0.008 | 6.06 |
| Region | 0.296 | 0.050 | 0.015 | 10.84 |
| Residence | −0.072 | −0.047 | −0.003 | −2.51 |
| Explained CIX | | | 0.178 | 81.18 |
| Residual CIX | | | −0.038 | 18.12 |

HF: Health Facility, CIX: Concentration index

**Table 7. Decomposition of CIX for measuring contributions of explanatory variables for socioeconomic inequalities in utilizing recommended ANC services (using administrative region as ranking variable)in Ethiopia from 2011–2019.**

| Variables | Elasticity | CIX | Contribution to overall [CIX = 0.07, $P < 0.0001$] | |
|---|---|---|---|---|
| | | | Absolute contribution | Percentage contribution |
| Mother's age | 0.333 | 0.004 | 0.001 | 2.13 |
| Mother education | 0.161 | 0.106 | 0.017 | 25.33 |
| Husband education | 0.037 | 0.066 | 0.002 | 3.68 |
| Wealth status | 0.353 | 0.017 | 0.006 | 8.86 |
| Distance of HF | 0.242 | −0.006 | −0.001 | −2.13 |
| Birth order | −0.166 | −0.009 | 0.001 | 2.15 |
| Wife beating | 0.050 | 0.011 | 0.001 | 0.85 |
| Region | 0.296 | 0.257 | 0.076 | 113.73 |
| Residence | 0.071 | 0.008 | −0.001 | −0.87 |
| Explained CIX | | | **0.102** | **153.73** |
| Residual CIX | | | **−0.032** | **−53.73** |

HF: Health Facility, CIX: Concentration index

budget on subsistence needs resulting in difficult trade-offs regarding spending on education and health care [37–39]. Health policymakers should prioritize efforts to reduce these inequalities and ensure equitable access to recommended ANC services utilization for all socioeconomic groups in Ethiopia.

## Policy implication of the study

Efforts should be made to address the underlying determinants of socioeconomic inequalities in recommended ANC services utilization such as wealth disparities, residential, educational, and regional disparities among mothers. The study suggested the need for development and implementation of targeted interventions to promote equitable access to recommended ANC services utilization in Ethiopia.

**Table 8. Decomposition of CIX for measuring contributions of explanatory variables for socioeconomic inequalities in utilizing recommended ANC services (using residence as ranking variable)in Ethiopia from 2011–2019.**

| Variables | Elasticity | CIX | Contribution to overall [CIX = −0.11, $P < 0.0001$] | |
|---|---|---|---|---|
| | | | Absolute contribution | Percentage contribution |
| Mother's age | 0.333 | 0.007 | 0.002 | −2.05 |
| Mother education | 0.161 | −0.219 | −0.035 | 31.61 |
| Husband education | 0.037 | −0.188 | −0.007 | 6.28 |
| Wealth status | 0.353 | −0.109 | −0.038 | 34.40 |
| Distance of HF | 0.242 | −0.050 | −0.012 | 10.92 |
| Birth order | −0.166 | 0.117 | −0.019 | 17.36 |
| Wife beating | 0.050 | −0.150 | −0.007 | 6.67 |
| Region | 0.296 | −0.037 | −0.011 | 9.82 |
| Residence | 0.072 | 0.104 | 0.008 | −6.73 |
| Explained CIX | | | −0.119 | 108.28 |
| Residual CIX | | | 0.009 | −8.28 |

HF: Health Facility, CIX: Concentration index

## Limitations of the study

This study utilized secondary data collected in 201, 2016, and 2019, which may not reflect the current status of recommended ANC services utilization in Ethiopia.

## Conclusion

Recommended ANC services utilization was lower than the national target and other study findings, even though it increased slightly from 2011 to 2019. Moreover, mothers' age and education, husband education, distance of health facilities, wealth status of the household, birth order and wife beating were significantly determining recommended ANC services utilization in Ethiopia. The study also revealed that there is a significant disparities existed across various socioeconomic groups in recommended ANC services utilization in Ethiopia.

## Supporting information

**S1 file**. **Excel dataset.**
(XLS)

## AcknowledgmentAcknowledgment

The authors are honored to appreciate Demographic and Health Surveys Program for providing EDHS datasets with authorization letter.

## Author contributions

**Conceptualization:** Yawkal Tsega, Gebeyehu Tsega, Chad Stecher.

**Data curation:** Yawkal Tsega, Abel Endawkie, Yeshimebet Ali Dawed.

**Formal analysis:** Yawkal Tsega, Chad Stecher.

**Funding acquisition:** Yawkal Tsega.

**Investigation:** Yawkal Tsega, Yeshimebet Ali Dawed, Chad Stecher.

**Methodology:** Yawkal Tsega, Abel Endawkie, Gebeyehu Tsega, Asnakew Molla Mekonen, Yeshimebet Ali Dawed, Chad Stecher.

**Project administration:** Yawkal Tsega.

**Resources:** Yawkal Tsega.

**Software:** Yawkal Tsega, Abel Endawkie.

**Supervision:** Gebeyehu Tsega, Asnakew Molla Mekonen, Yeshimebet Ali Dawed, Chad Stecher.

**Validation:** Gebeyehu Tsega, Yeshimebet Ali Dawed, Chad Stecher.

**Visualization:** Yawkal Tsega.

**Writing – original draft:** Yawkal Tsega, Abel Endawkie, Gebeyehu Tsega.

**Writing – review & editing:** Yawkal Tsega, Abel Endawkie, Gebeyehu Tsega, Asnakew Molla Mekonen, Yeshimebet Ali Dawed, Chad Stecher.

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
