## [Decision Letter · Decision Letter 0]

9 Oct 2024

PONE-D-24-13694Trends and Socioeconomic Inequality in Recommended Antenatal Care (ANC) Service Utilization in Ethiopia:  A Decomposition Analysis Using Ethiopian Nationwide Demographic Health Survey Data from 2011-2019PLOS ONE

Dear Dr. Tsega,

Thank you for submitting your manuscript to PLOS ONE. After careful consideration, we feel that it has merit but does not fully meet PLOS ONE’s publication criteria as it currently stands. Therefore, we invite you to submit a revised version of the manuscript that addresses the points raised during the review process.

 Please submit your revised manuscript by Nov 22 2024 11:59PM. If you will need more time than this to complete your revisions, please reply to this message or contact the journal office at plosone@plos.org . Please include the following items when submitting your revised manuscript:

We look forward to receiving your revised manuscript.

Kind regards,

Joanna Tindall, PhD

Staff Editor

PLOS ONE

Journal Requirements:

2. We note that your Data Availability Statement is currently as follows: “All relevant data are within the manuscript and in Supporting Information files.”

Please confirm at this time whether or not your submission contains all raw data required to replicate the results of your study. Authors must share the “minimal data set” for their submission. PLOS defines the minimal data set to consist of the data required to replicate all study findings reported in the article, as well as related metadata and methods (https://journals.plos.org/plosone/s/data-availability#loc-minimal-data-set-definition). For example, authors should submit the following data: - The values behind the means, standard deviations and other measures reported; - The values used to build graphs; - The points extracted from images for analysis. Authors do not need to submit their entire data set if only a portion of the data was used in the reported study. If your submission does not contain these data, please either upload them as Supporting Information files or deposit them to a stable, public repository and provide us with the relevant URLs, DOIs, or accession numbers. For a list of recommended repositories, please see https://journals.plos.org/plosone/s/recommended-repositories. If there are ethical or legal restrictions on sharing a de-identified data set, please explain them in detail (e.g., data contain potentially sensitive information, data are owned by a third-party organization, etc.) and who has imposed them (e.g., an ethics committee). Please also provide contact information for a data access committee, ethics committee, or other institutional body to which data requests may be sent. If data are owned by a third party, please indicate how others may request data access.

Additional Editor Comments:

Please note that we have only been able to secure a single reviewer to assess your manuscript. We are issuing a decision on your manuscript at this point to prevent further delays in the evaluation of your manuscript. Please be aware that the editor who handles your revised manuscript might find it necessary to invite additional reviewers to assess this work once the revised manuscript is submitted. However, we will aim to proceed on the basis of this single review if possible. 

Reviewers' comments:

Reviewer's Responses to Questions

**Comments to the Author**

1. Is the manuscript technically sound, and do the data support the conclusions?

Reviewer #1: Yes

2. Has the statistical analysis been performed appropriately and rigorously? 

Reviewer #1: Yes

3. Have the authors made all data underlying the findings in their manuscript fully available?

Reviewer #1: Yes

4. Is the manuscript presented in an intelligible fashion and written in standard English?

Reviewer #1: Yes

5. Review Comments to the Author

Reviewer #1: This manuscript analyzed the patterns and socioeconomic disparities in the utilization of recommended antenatal care (ANC) services in Ethiopia. It used nationwide demographic health survey data from 2011 to 2019. In general, the research is solid, carefully designed, and the paper is skillfully written and well-structured. Having said that, I strongly urge the authors to focus on the following few issues:

1. Some attention is needed for lines 207-208 by the authors.

2. Line 232: The authors wroye: (Error! Reference source not found.). What is the meaning of this statement?

3. Unlike the Tables, all Figures clearly require significant revisions to enhance their quality.

4. Editing is needed for the cell in Table 6, specifically in column 3 and row 4.

5. While the discussion is acceptable, it could be enhanced by including more comparisons with other studies conducted in Ethiopia or the surrounding region that share similar conditions.

6. While the discussion is acceptable, it could be enhanced by including more comparisons with other studies conducted in Ethiopia or the surrounding region that share similar conditions.

6. PLOS authors have the option to publish the peer review history of their article (what does this mean? ). If published, this will include your full peer review and any attached files.

**Do you want your identity to be public for this peer review?** For information about this choice, including consent withdrawal, please see our Privacy Policy .

Reviewer #1: No

---

## [Author Response · Author response to Decision Letter 0]

15 Oct 2024

Date: October 16, 2024

To: PLOS ONE

Subject: Submission of Revised Manuscript

Dear Editor and Reviewer,

We, the authors, are deeply appreciative of your response to our manuscript # PONE-D-24-13694, titled "Trends and Socioeconomic Inequality in Recommended Antenatal Care Service Utilization in Ethiopia: A Decomposition Analysis Using Ethiopian Nationwide Demographic Health Surveys Data from 2011-2019". Your valuable comments, and expert suggestions are instrumental to enhance the quality of our work.

We have not used retracted references and have meticulously revised and updated the manuscript taking the editor and reviewer’s comments and suggestions into consideration. We are eager to publish the manuscript in your reputable journal to reach a relevant audience and influence policy changes aimed at improving antenatal care services in Ethiopia and similar contexts.

Once again, we are grateful for your comments and suggestions.

Best regards,

Corresponding author: Yawkal Tsega

Email: yawkaltsega@gmail.com

Mobile: +251933559351

Response to Editors comment

Comment 1: Please note that we have only been able to secure a single reviewer to assess your manuscript. We are issuing a decision on your manuscript at this point to prevent further delays in the evaluation of your manuscript. Please be aware that the editor who handles your revised manuscript might find it necessary to invite additional reviewers to assess this work once the revised manuscript is submitted. However, we will aim to proceed on the basis of this single review if possible.

Authors’ response: Dear Editor, thank you for your update regarding the review process. We appreciate the efforts made to secure a reviewer for our manuscript. We revised the manuscript and proofread for grammatical errors and submitted the revised manuscript with rebuttal letter and point by point responses.

Response to Reviewer 1 comments

Comment 1: This manuscript analyzed the patterns and socioeconomic disparities in the utilization of recommended antenatal care (ANC) services in Ethiopia. It used nationwide demographic health survey data from 2011 to 2019. In general, the research is solid, carefully designed, and the paper is skillfully written and well-structured. Having said that, I strongly urge the authors to focus on the following few issues:

Authors’ response: Dear reviewer 1, thank you for reviewing our manuscript and your encouraging kind words. We addressed all the comments, suggestions and recommendations in the revised manuscript as per your comments and suggestions.

Comment 2: Some attention is needed for lines 207-208 by the authors.

Authors’ response: Thank you for your comments. We addressed the comment in the revised manuscript.

Comment 3: Line 232: The authors wrote: (Error! Reference source not found.). What is the meaning of this statement?

Authors’ response: Thank you for bringing this editorial error into our attention. We revised statement in the revised version of the manuscript. It was due to the error of Table citation.

Comment 4: Unlike the Tables, all Figures clearly require significant revisions to enhance their quality.

Authors’ response: We are grateful for your comments and suggestions. We tried to increase the quality of figures to increase the visibility.

Comment 5: Editing is needed for the cell in Table 6, specifically in column 3 and row 4.

Authors’ response: Thank you for your suggestion. We revised and proofread table 6 and the overall contents of the manuscript.

Comment 6: While the discussion is acceptable, it could be enhanced by including more comparisons with other studies conducted in Ethiopia or the surrounding region that share similar conditions.

Authors’ response: Thank you for your valuable feedback. We appreciate the suggestion to include more comparisons with other studies conducted in Ethiopia or the surrounding region. We incorporated relevant studies to enhance the discussion and provide a more comprehensive analysis of the conditions and findings.

---

## [Decision Letter · Decision Letter 1]

5 Dec 2024

PONE-D-24-13694R1Trends and socioeconomic inequality in recommended Antenatal Care Service utilization in Ethiopia: A Decomposition Analysis using Ethiopian Nationwide Demographic Health Surveys from 2011-2019PLOS ONE

Dear Dr. Tsega,

Thank you for submitting your manuscript to PLOS ONE. After careful consideration, we feel that it has merit but does not fully meet PLOS ONE’s publication criteria as it currently stands. Therefore, we invite you to submit a revised version of the manuscript that addresses the points raised during the review process.

We look forward to receiving your revised manuscript.

Kind regards,

Ranjan Kumar Prusty, Ph.D.

Academic Editor

PLOS ONE

Reviewers' comments:

Reviewer's Responses to Questions

**Comments to the Author**

1. If the authors have adequately addressed your comments raised in a previous round of review and you feel that this manuscript is now acceptable for publication, you may indicate that here to bypass the “Comments to the Author” section, enter your conflict of interest statement in the “Confidential to Editor” section, and submit your "Accept" recommendation.

Reviewer #2: (No Response)

Reviewer #3: (No Response)

2. Is the manuscript technically sound, and do the data support the conclusions?

Reviewer #2: Yes

Reviewer #3: No

3. Has the statistical analysis been performed appropriately and rigorously? 

Reviewer #2: Yes

Reviewer #3: I Don't Know

4. Have the authors made all data underlying the findings in their manuscript fully available?

Reviewer #2: Yes

Reviewer #3: No

5. Is the manuscript presented in an intelligible fashion and written in standard English?

Reviewer #2: Yes

Reviewer #3: No

6. Review Comments to the Author

Reviewer #2: I have few comments

1. In the methodology, The authors have kept individual level socioeconomic status and community or macro indicators such as HDI together, it will be good if they present it with separate subtitle.

2. Further, Mother’s current marital status is described as currently unmarried, currently married. It will be good if Currently unmarried presented as widowed/separated

3. In table 3, for the Family size, in the COR the authors have kept ≥7 as reference and in the adjusted model, the authors have kept 1-4 as reference, why?. Its better to follow same

Reviewer #3: Reviewer’s report:

Manuscript Number: PONE-D-24-13694R1

Article Title: Trends and socioeconomic inequality in recommended Antenatal Care Service utilization in Ethiopia: A Decomposition Analysis using Ethiopian Nationwide Demographic Health Surveys from 2011-2019

Overview:

This paper, submitted to PLOS ONE, has focussed on an important public health issue in LICs, socioeconomic inequality in utilization of antenatal care services among women. The study used data from three waves of Ethiopian DHS conducted in years 2011, 2016, and 2019. Bivariate and multivariate logistic regression models were applied to find out the determinants of recommended ANC services utilization among Ethiopian mothers. Further, decomposition of concentration index was carried out to measure contributions of studied variables in socioeconomic inequalities.

Finding highlights that coverage of recommended ANC services increased from 29.96% in 2011 to 44.70% in 2019. Several socio-demographic factors such as mother’s age and education, household wealth status, distance of mothers’, place to the nearest health facility, among others were significantly associated with utilization of recommended ANC services. The decomposition of concentration indices revealed household wealth status as the largest contributor (~67%) to the socioeconomic inequality that exist in utilization of antenatal care services among Ethiopian women.

The current version of the paper does not meet the publication quality. The manuscript should be thoroughly revised by carefully addressing a number of shortcomings, ranging from its grammatical errors to statistical analyses. Authors may consider my comments below while revising the manuscript.

I appreciate the opportunity to review this work and hope that authors would find my comments useful.

Observations:

— The information presented in the ‘Introduction’ section is very generic. The research problem to be addressed (in this study) should be described adequately and with an appropriate background, which is missing.

— Where is the literature review? Authors should add at least one paragraph on review of relevant literature to the ‘Introduction’ section.

— Next is research gap. Authors should describe whether there were any research gaps in the existing literature and thereafter, the need for this study should be mentioned.

— The methodology used in the study should be introduced to the readers in the ‘Introduction’ section only.

— Line 80: Authors may consider renaming of the heading ‘Data and Methods’.

— Line 89: What is ‘source population’?.

— Line 90-91: It should be mentioned clearly about who were the study population (e.g., pregnant women or women of reproductive age).

— How do you define ‘Recommended ANC Services’? Please disclose it to readers while defining outcome variables.

— Line 115: Remove the statement “and it is also more practical to implement”

— Line 199-201, 202-203: Not clear, what authors want to state.

— Line 218: What does it mean by ‘candidates’?

— Line 401-403: Not clear, what authors want to state.

— The study should be organised appropriately ensuring that it is not taxing the readers’ attention.

— Finally, this manuscript needs thorough English language editing to make it understandable to the readers.

7. PLOS authors have the option to publish the peer review history of their article (what does this mean? ). If published, this will include your full peer review and any attached files.

**Do you want your identity to be public for this peer review?** For information about this choice, including consent withdrawal, please see our Privacy Policy .

Reviewer #2: No

Reviewer #3: No

---

## [Author Response · Author response to Decision Letter 1]

8 Dec 2024

Date: 08 December 2024

To: PLOS ONE

Subject: Submission of Revised Manuscript

Dear Editor and Reviewers,

We, the authors, are deeply appreciative of your response to our manuscript # PONE-D-24-13694R1, titled " Trends and socioeconomic inequalities of recommended antenatal care services utilization in Ethiopia: A Decomposition analysis using Ethiopian nationwide Demographic Health Surveys 2011-2019". Your valuable comments, and expert suggestions are instrumental to enhance the quality of our manuscript.

We have revised the manuscript taking the reviewers’ comments and suggestions into consideration. We are eager to publish the manuscript in your reputable journal to reach a relevant audience and influence policy changes aimed at improving antenatal care services in Ethiopia and other similar contexts.

Once again, we are grateful for the editor and reviewers for their expert comments, recommendations, and suggestions.

Best regards,

Yawkal Tsega

On behalf of the authors

Response to Editors comment

Comment 1: Thank you for submitting your manuscript to PLOS ONE. After careful consideration, we feel that it has merit but does not fully meet PLOS ONE’s publication criteria as it currently stands. Therefore, we invite you to submit a revised version of the manuscript that addresses the points raised during the review process.

Authors’ response: Dear Editor, Thank you so much for your update and time in evaluating our manuscript. We revised and tried to address all the comments and suggestions provided by the reviewers and submitted the revised manuscript.

Response to Reviewer 2 comments

Comment 1: In the methodology, the authors have kept individual level socioeconomic status and community or macro indicators such as HDI together, it will be good if they present it with separate subtitle.

Authors’ response: Dear Reviewer 2, we are grateful for your insightful comments and suggestions. Your comments and suggestions have been instrumental in enhancing the quality of our work. Based on your feedback, we have separated the individual and community-level factors in the revised manuscript.

Comment 2: Further, Mother’s current marital status is described as currently unmarried, currently married. It will be good if currently unmarried presented as widowed/separated

Authors’ response: Thank you for your insightful recommendations. We have clarified the meaning of "currently unmarried" (single/divorced/widowed) in the footnote of the table in the revised manuscript, taking your suggestions into consideration.

Comment 3: In table 3, for the Family size, in the COR the authors have kept ≥7 as reference and in the adjusted model, the authors have kept 1-4 as reference, why? It’s better to follow same

Authors’ response: Thank you so much for detailed comments of the manuscript. It was due to editorial errors. We have corrected it in the revised manuscript.

Response to Reviewer 3 comments

Comment 1: Overview: This paper, submitted to PLOS ONE, has focused on an important public health issue in LICs, socioeconomic inequality in utilization of antenatal care services among women. The study used data from three waves of Ethiopian DHS conducted in years 2011, 2016, and 2019. Bivariate and multivariate logistic regression models were applied to find out the determinants of recommended ANC services utilization among Ethiopian mothers. Further, decomposition of concentration index was carried out to measure contributions of studied variables in socioeconomic inequalities.

Finding highlights that coverage of recommended ANC services increased from 29.96% in 2011 to 44.70% in 2019. Several socio-demographic factors such as mother’s age and education, household wealth status, distance of mothers’, place to the nearest health facility, among others were significantly associated with utilization of recommended ANC services. The decomposition of concentration indices revealed household wealth status as the largest contributor (~67%) to the socioeconomic inequality that exist in utilization of antenatal care services among Ethiopian women.

Authors’ response: Dear reviewer 3, we appreciate for your time, expert comments, suggestions, and recommendations. Your detailed comments have played significant role in the quality improvement of our work. We have tried to address all your suggestions, and comments in the revised manuscript.

Comment 2: The current version of the paper does not meet the publication quality. The manuscript should be thoroughly revised by carefully addressing a number of shortcomings, ranging from its grammatical errors to statistical analyses. Authors may consider my comments below while revising the manuscript. I appreciate the opportunity to review this work and hope that authors would find my comments useful.

Authors’ response: Your comments are highly valuable for the quality of our work, and we appreciate them. We have repeatedly proofread the manuscript and addressed the grammatical errors, punctuation errors, and logical flow.

Comment 3: The information presented in the ‘Introduction’ section is very generic. The research problem to be addressed (in this study) should be described adequately and with an appropriate background, which is missing.

Authors’ response: Thank you for your deep and detailed comments. We have revised the introduction section to better explain the research problems, existing literature, and the gap this research fills, taking your suggestions into consideration.

Comment 4: Where is the literature review? Authors should add at least one paragraph on review of relevant literature to the ‘Introduction’ section.

Authors’ response: We are grateful for your recommendation. We added literatures in the revised manuscript tried to improve the introduction part in the revised manuscript.

Comment 5: Next is research gap. Authors should describe whether there were any research gaps in the existing literature and thereafter, the need for this study should be mentioned.

Authors’ response: We have thoroughly revised the introduction section in the revised manuscript. We have explained the rationale for the current study by highlighting the gaps in the existing literature.

Comment 6: The methodology used in the study should be introduced to the readers in the ‘Introduction’ section only.

Authors’ response: We summarized the methods used in the study in the revised manuscript.

Comment 7: Line 80: Authors may consider renaming of the heading ‘Data and Methods’.

Authors’ response: We are grateful for your expert comments and suggestions. We clearly wrote the headings according to PLOS ONE manuscript submission guideline.

Comment 8: Line 89: What is ‘source population’?

Authors’ response: Dear Reviewer 3, we are grateful for your expert comments and suggestions. We clearly stated the source and study populations in the revised manuscript. “All pregnant women five years preceding each respective survey year were the source populations, and all pregnant women five years preceding each survey year in the selected EAs were the study populations”.

Comment 9: Line 90-91: It should be mentioned clearly about who were the study population (e.g., pregnant women or women of reproductive age).

Authors’ response: We appreciate your expert comments and suggestions. We clearly stated the source and study populations in the revised manuscript. “All pregnant women five years preceding each respective survey year were the source populations, and all pregnant women five years preceding each survey year in the selected EAs were the study populations”.

Comment 10: How do you define ‘Recommended ANC Services’? Please disclose it to readers while defining outcome variables.

Authors’ response: Thank you so much. Recommended ANC services utilization was considered “if the mother attended her first ANC visit within first trimester of her pregnancy and if she completed at least four ANC visits”. We operationalized the outcome variable, recommended ANC services utilization in the revised manuscript.

Comment 11: Line 115: Remove the statement “and it is also more practical to implement”

Authors’ response: We revised the statement in the revised manuscript as per your suggestions.

Comment 12: Line 199-201, 202-203: Not clear, what authors want to state.

Authors’ response: From line 199-203, we stated the ethical approval. We revised the section and tried to make it clearer in the revised manuscript.

Comment 13: Line 218: What does it mean by ‘candidates’?

Authors’ response: Thank you so much for your question. We revised the statement in the revised manuscript. It was just to explain the eligible variables for multivariable logistic regression.

Comment 14: Line 401-403: Not clear, what authors want to state.

Authors’ response: We greatly appreciate your insightful and expert comments. In lines 401-403, we intended to explain the policy implications of the study findings. However, we have revised the paragraph to be more specific and clearer in the revised manuscript, taking your suggestions into consideration.

Comment 15: The study should be organized appropriately ensuring that it is not taxing the readers’ attention.

Authors’ response: We conducted a thorough proofread of the whole manuscript word by word and tried to enhance the organization and quality of the manuscript in the current revised manuscript.

Comment 16: Finally, this manuscript needs thorough English language editing to make it understandable to the readers.

Authors’ response: Dear Reviewer 3, we are very grateful for your insightful and expert comments, suggestions, and recommendations. We have gladly accepted your suggestions and conducted a thorough proofread of the entire manuscript, word by word, to enhance its organization and quality in the revised version.

---

## [Editor Report · Decision Letter 2]

15 Jan 2025

Trends and socioeconomic inequalities of recommended antenatal care services utilization in Ethiopia: A Decomposition analysis using Ethiopian nationwide Demographic Health Surveys 2011-2019

PONE-D-24-13694R2

Dear Dr. Tsega,

We’re pleased to inform you that your manuscript has been judged scientifically suitable for publication and will be formally accepted for publication once it meets all outstanding technical requirements.

Kind regards,

Ranjan Kumar Prusty, Ph.D.

Academic Editor

PLOS ONE
---

## [Editor Report · Acceptance letter]

PONE-D-24-13694R2

PLOS ONE

Dear Dr. Tsega,

I'm pleased to inform you that your manuscript has been deemed suitable for publication in PLOS ONE. Congratulations! Your manuscript is now being handed over to our production team.

Kind regards,

on behalf of

Dr. Ranjan Kumar Prusty

Academic Editor

PLOS ONE